# Phage Therapy for Crops: Concepts, Experimental and Bioinformatics Approaches to Direct Its Application

**DOI:** 10.3390/ijms24010325

**Published:** 2022-12-25

**Authors:** José Luis Villalpando-Aguilar, Gilberto Matos-Pech, Itzel López-Rosas, Hugo Gildardo Castelán-Sánchez, Fulgencio Alatorre-Cobos

**Affiliations:** 1Unidad Académica Yucatán, Instituto de Investigaciones en Matemáticas Aplicadas y en Sistemas, Universidad Nacional Autónoma de México, Mérida 97302, Mexico; 2El Colegio de la Frontera Sur, Unidad Campeche, Lerma, Campeche 24500, Mexico; 3Consejo Nacional de Ciencia y Tecnología—Colegio de Postgraduados Campus Campeche, Champotón, Campeche 24450, Mexico; 4Programa de Investigadoras e Investigadores por Mexico, Grupo de Genómica y Dinámica Evolutiva de Microorganismos Emergentes, Consejo Nacional de Ciencia y Tecnología, Benito Juárez, Ciudad de México 03940, Mexico; 5Consejo Nacional de Ciencia y Tecnología—Unidad de Bioquímica y Biología Molecular de Plantas, Centro de Investigación Científica de Yucatán, Mérida 97205, Mexico

**Keywords:** bacteriophages, genomics, phage therapy, crop, bioinformatics, phage genome, biocontrol

## Abstract

Phage therapy consists of applying bacteriophages, whose natural function is to kill specific bacteria. Bacteriophages are safe, evolve together with their host, and are environmentally friendly. At present, the indiscriminate use of antibiotics and salt minerals (Zn^2+^ or Cu^2+^) has caused the emergence of resistant strains that infect crops, causing difficulties and loss of food production. Phage therapy is an alternative that has shown positive results and can improve the treatments available for agriculture. However, the success of phage therapy depends on finding effective bacteriophages. This review focused on describing the potential, up to now, of applying phage therapy as an alternative treatment against bacterial diseases, with sustainable improvement in food production. We described the current isolation techniques, characterization, detection, and selection of lytic phages, highlighting the importance of complementary studies using genome analysis of the phage and its host. Finally, among these studies, we concentrated on the most relevant bacteriophages used for biocontrol of *Pseudomonas* spp., *Xanthomonas* spp., *Pectobacterium* spp., *Ralstonia* spp., *Burkholderia* spp., *Dickeya* spp., *Clavibacter michiganensis*, and *Agrobacterium tumefaciens* as agents that cause damage to crops, and affect food production around the world.

## 1. Introduction

Earth is home to a great diversity of bacteriophages, which destroy between 4% and 50% of bacteria, thus contributing to natural bacterial control [1]. 

In 1917, Félix Hubert d’Herelle used the term “bacteriophages” for the first time, focusing on their application for bacterial biocontrol [2]. In the case of phage therapy in plants, in 1924, Mallman and Hermstreet discovered that a filtrate of decomposed cabbage inhibited the growth of *Xanthomonas campestris* pv. *campestris*, responsible for cabbage rot disease [3].

In 1926, Moore described experiments that showed successful biocontrol of *Erwinia carotovora* subsp. *atroseptica* (currently *Pectobacterium carotova* subsp. *atroseptica*), the causative agent of potato tuber rot, and promoted phage therapy as an alternative for disease biocontrol in crops [4]. However, phage therapy was overshadowed by other bacterial disease control alternatives. During 1928–1940, the British scientist Alexander Fleming discovered penicillin, giving rise to the antibiotics era, with a concomitant decreasing interest in applying phages for bacterial control [5]. In 1975, Frederick Williams Twort studied a “bacterial lytic factor” identified as a compound capable of generating bacterial lysis [6].

Since then, antibiotics and mineral salts (Zn^2+^ or Cu^2+^) have been indiscriminately employed as bacterial treatments for crops, and have generated resistant strains, thus promoting research on biocontrol alternatives for super-resistant bacteria. One option is phage therapy, which uses bacteriophages as agents that kill bacteria. The advantages are that bacteriophages evolve together with their host; due to their great diversity, it is increasingly possible to find phages for specific diseases; and they are safe and compatible with the environment [7,8].

Therefore, the success of phage therapy is limited by the selection process, in which a high reproduction rate (PFU/mL) and lytic capacity are sought. It is more likely to isolate a lysogenic phage than a lytic phage. Therefore, it is essential to involve modern approaches, such as studying the genetic material of the host (bacteria) and the control agent (phage), which, combined, will increase the possibility of selecting lytic phages for use as bacterial biocontrol agents and reduce the time it takes to do so [9]. 

It should be noted that it is possible to predict structural characteristics, infection mechanisms, and lytic capacity from phage genomes. On the other hand, bacterial genomes allow the selection of stable hosts for phage propagation, infection susceptibility, and receptor prediction as means of interaction for successful infection. Therefore, complementing experimental strategies with in silico analysis of bacteria and phages can enhance phage therapy for agricultural cultures [10,11].

In this context, bacteriophages are applied as an alternative treatment for bacterial agricultural diseases, and they have shown the ability to reduce *Pseudomonas* spp., *Xanthomonas* spp., *Pectobacterium* spp., *Ralstonia* spp., *Clavibacter michiganensis*, and *Agrobacterium tumefaciens* [12]. In the USA, products approved by the FDA for use in agriculture, such as AgriPhage, have shown positive results in the biocontrol of pepper spot, speck, and tomato canker [13].

In this review, we summarized the application of phage therapy on crops by describing the structure of phages that infect phytobacteria; strategies for isolation, detection, and selection; and the study of phage and bacterial genomes through bioinformatics approaches. Finally, we highlight the most relevant studies that have applied phage therapy for biocontrol of bacteria that commonly infect crops of agricultural importance.

## 2. Bacteriophages Used for Biocontrol of Crop Diseases 

The class *Caudoviricetes* is the most studied among the phage groups, as it is the order in which phages for biocontrol of agricultural crop diseases are found. Its members have an icosahedral capsid and a tail containing trimeric protein fibers in the lower part that serves as a receptor for interaction with the host cell, with stability in L morphology [14]. The class *Caudoviricetes* include phages according to the tail shape: the myovirus includes phages with long and contractile tails that stand out, the podovirus have short non-contractile tails, and the siphovirus have long and flexible tails (Figure 1) [14,15]. One of the most studied phages is a tailed phage with a dsDNA genome which has three main parts in its structure: a capsid, a tail, and an absorption apparatus. In the assembly process, the capsid of bacteriophages has a structure called a procapsid. Scaffolding proteins have the function of supervising other subunits of the main capsid to facilitate the formation of the icosahedral procapsid, where the dsDNA will be stored. The mechanism of change from procapsid to capsid within the genome is called the ripening process.

The organization of phage tails is associated with the type of bacteriophage; for instance, siphovirus have long and flexible tails, podovirus have short tails with adhesive properties, and myovirus have long, rigid, contractile tails that shape the rigid internal tube and external contractile sheath. On the other hand, many phages that can infect gram-negative bacteria have an absorption apparatus, which is an oligomeric ring formed by proteins from the distal tail attached to the tailed tube’s last ring. It has the function of recognizing and connecting with receptor-binding proteins [17].

The tectivirus genus structure consists of a rigid protein capsid containing a thick lipoprotein and a flexible vesicle and dsDNA, which gives these phages the ability to infect both gram-positive bacteria, such as the betatectivirus genus, and gram-negative bacteria with the alphatectivirus and PRD1 genus [18], which is important, as gram-negative bacteria cause the most problems in economically important crops [19]. The corticovirus also have lipid layers in the protein capsid; they infect gram-negative *Pseudoalteromonas* spp. and are associated with a PM2 capsid architecture with fold trimeric proteins containing two β-barrels forming hexagonal capsomeres [20].

In the case of archaeal viruses, they can be divided into two groups: In the first group, the relationships between the morphologic and genetic aspects of these viruses are unique. The second group has clear genetic and structural similarities to bacteriophages and eukaryotic viruses. The majority of archaea viruses have linear or circular dsDNA genomes, while only two families present single DNA genomes. No RNA viruses have been isolated, but they have been detected in metagenomic studies [21].

Currently, studies are focused on the structure and assembly of archaeal viruses, which requires characterization of unexplored host phyla to increase our knowledge of archaeal virus diversity [22].

Proteobacteria are mainly gram-negative and cause many diseases in agricultural crops. From a structural point of view, some examples show how the structure of bacteriophage is related to bacterial species that cause damage to agricultural crops. For instance, *Xanthomonas* spp. are infected by phage Xaj2 and phage Xf2. Phage RSM3 infects *Ralstonia solanacearum*. *Pseudomonas syringae* is infected by like phage phobos or phage MR1-MR18, classified in podovirus or myovirus, and cystovirus as the phage phi6 or phi8. The most studied are myovirus phage T4 and levivirus as phage MS2, which infect *E. coli* (Figure 1) [23,24,25,26]. 

## 3. Bacteriophage Classification

The International Committee on Taxonomy of Viruses (ICTV) has established the official taxonomy. Historically, in the past, viruses were classified based on several criteria, such as propagation characteristics in cell culture, virion morphology, serology, nucleic acid sequence, host range, pathogenicity, and epidemiology or epizootiology [27]. Recently, new proposals have been made to improve virus classification because there is no standardized and automatic universally accepted virus classification. Technological advances in sequencing, and especially metagenomics projects, have increased the available phage genomes and have since been used for phage genomes as a criterion for classification [28]. Essentially, Turner et al. described the abolition of the order *Caudovirales* and the families *Myoviridae*, *Podoviridae*, and *Siphoviridae* [29]. In bacteriophages, the members of the largest order (Caudovirales) were assigned to the class *Caudoviricetes*. The old families (*Myoviridae, Podoviridae*, and *Siphoviridae*) were replaced by the families *Strabovoridae*, *Drexlerviridae*, and *Autographiviridae*, which have a high similarity to the old families [30,31].

In addition, the order *Tubulavirales*, which includes the phages of the family *Inoviridae*, was divided into two families, *Inoviridae* and *Plectroviridae* [32,33]. In the family *Microviridae*, additional subfamilies beyond the existing *Gokushovirinae* and *Bullavirinae* have been proposed, namely the subfamilies *Alpavirinae*, *Stokavirinae*, *Aravirinae,* and *Pichovirinae* based on virome data. Finally, the family *Leviviridae*, which described a comprehensive identification of ssRNA based on computational approaches, used the phage genome [33,34,35,36].

Currently, Walker et al. reported recent changes in viral taxonomy ratified by ICTV; approximately 60% of all species names have been reordered. For bacteriophages, classification improves the taxonomy of monophyletic genome-based families [37].

This review showed a collection of phages used for the therapy of various crops. Nevertheless, the application of genome-based taxonomic classification is limited by its availability. However, we have shown in this review that not all genome phages are available because we used the old taxonomy in describing the different phages cited here. In further studies, we will perform computational approaches to update the taxonomy for various phages cited in this review with available genomes.

## 4. Bacteriophage Infection Mechanisms

Bacteriophages are capable of reproducing by two biological strategies to perpetuate their genetic material. One strategy is the lytic cycle, which is the most aggressive, since it kills the host cell. The first step consists of fixation: the bacteriophage binds to the host cell through a ligand–receptor interaction [38]. This interaction is so specific that it allows differentiation between gram-negative and gram-positive hosts [39].

The first interactions are made by the virus to specific receptors, such as lipopolysaccharides (LPSs) or outer membrane proteins. The binding of two, and even up to six fibers to the receptor sends a signal to the base of the base plate that promotes a conformational change, leading to contraction of the fibers, which readies the bacteriophage into an injection position. Mainly, the stem pierces the membrane, generating a channel that allows the injection of genetic material. This perforation is supported by endopeptidases (N-acetylmuramyl-L-alanine amidase, lysozymes, transglycosylases), which commonly degrade peptidoglycan; these enzymes are part of the fiber structure of bacteriophages [38]. With the genetic material introduced into the bacterium, the host’s replication, transcription, and translation machinery are hijacked to generate the genetic material and to synthesize and obtain viral proteins. First, protein messengers are synthesized to stabilize and protect DNA or RNA molecules from degradation. In DNA viruses, the genetic material is used directly as a template for transcribing genes that code for structural proteins. In RNA viruses, a reverse transcription step is required to achieve the genetic material to be replicated. The genes that encode structural proteins are transcribed, and as soon as the structural proteins are generated, the copies of genetic material begin to assemble into the virion, stabilizing the DNA or RNA molecules with proteins. Then the capsid, the stem, and the fibers are assembled with the base plate. Additionally, lytic enzymes known as lysines are synthesized, which are encoded in the genetic material injected by the bacteriophage, and are the enzymes responsible for breaking the plasma membrane and allowing external liquid to enter the cytosol of the bacterium, which takes it to a point where the membrane breaks and releases the internal content, along with the virions, to start the lytic cycle again [38,39].

An important protein during the lytic cycle is the holin (hole-forming) protein, which exist in double-stranded DNA bacteriophages and control the length of the infectious cycle. Holins are small proteins that accumulate on the membrane and cause permeabilization. These proteins can be classified into three types according to their topology: class I with 95 residues that form three TMDs, class II with 65 to 95 residues that form two TMDs, and class III, with one TMD in the central region of the molecule [40,41].

The second method of reproduction and conservation of viral genetic material bacteriophages that have been developed and are used widely is the lysogenic cycle, which consists of the steps of fixation, injection of genetic material, and the lytic cycle. However, in this case, viral DNA is integrated into the bacterial chromosome and remains there in an inactive form as a prophage. This allows conservation of the bacteriophage sequence, which is replicated and transferred to the bacterial daughter cells through the bacterial chromosome DNA, where the virus is also duplicated. The genetic material is integrated through binding sites located on the bacterial chromosome by factors of the bacteriophage, and not by recombination or the integrated systems of the bacteria [38].

This strategy is more elegant and does not affect the viability of the host. In the literature, it is described as a strategy that seeks to preserve the host. Since bacteriophages are specific, it is believed that they sometimes use the lysogenic cycle to keep host cells alive, because if they carried out the lytic cycle all the time, they could exterminate their hosts and therefore would also be condemned to die [38].

## 5. Strategies for Bacteriophage Isolation from Plants

In general, the bacteriophage isolation process is as follows: First, the inoculum (phage) must be isolated, and the host bacteria must be identified; it is enough to inoculate a bacterial culture with a phage inoculum and incubate it to obtain a higher bacteriophage titer, which can be clarified by centrifugation or filtration [9].

A challenge is encountered with lytic phages, which is one of the limitations of phage therapy. The characteristics that a phage must have to be a candidate for phage therapy are lytic capacity, high progeny, and host specificity, which means it infects a single species of bacteria while leaving the rest of the microbiome intact [42]. However, in practice, it has been reported that phages can have more than one host and can attack groups of bacterial strains, which would enable solving diseases caused by a variety of bacterial strains by using a single phage with a broad spectrum of hosts. However, this is complicated when different species of bacteria cause disease; in this case, specific phages against specific bacterial strains can no longer solve the problem. A strategy that, in some cases, has made it possible to counteract bacterial growth is the use of a “phage cocktail”, which has shown promising results [19,43].

Therefore, a determining step in the success of phage therapy is isolation and characterization, and taking into consideration the type of sample and the host.

The primary isolation method, which was developed by Félix d’Herelle, consists of an enrichment process [44]. First, a sample of bacteria (host) is mixed with an environmental sample, and this must be close to the area of infection to be treated. In order to obtain phages against a specific disease, it is recommended to take a sample close to the infected site, either from the leaf (5 cm^2^), stem (2 cm), soil (150 g), or irrigation water (10 mL) of the infected plant. This is then mixed with the host bacterial culture [12,45]. Generally, soil samples have the highest concentration of phages. After the mixture of bacteria (host) with the environmental sample (phages) is prepared, there is an incubation period to 28–37 °C for approximately 16 to 18 h in a shaker (180 to 200 rpm/min). The application and selection of enrichment media will depend on the chosen host bacteria and the cell biomass that must be produced [46].

In general, the most common bacteria that cause plant diseases are *Pectobacterium, Pantoea* [47], *Agrobacterium* [48], *Pseudomonas* [49], *Ralstonia* [50], *Burkholderia* [51], A*cidovorax*, *Xanthomonas* [52], *Clavibacter*, *Streptomyces* [38], *Xylella* [39], *Spiroplasma*, and *Phytoplasma* [41]. The media used for the management of these species are Luria broth (LB), PEB medium, Lennox broth supplemented with calcium, nutrient broth (NB), semi-solid yeast extract agar (NYA), periwinkle wilt (PW) medium, and ATCC medium: 988 *Spiroplasma* SP-4 medium. Therefore, in the process of obtaining phages, it is necessary to characterize the host bacterial strain to obtain high titers and phage production [39].

Accordingly, bacteria or cell debris are removed from the culture by a physical method such as centrifugation or filtration to analyze the presence of phages. The identified phages are characterized to determine the desired properties for therapy based on their virulence capacity [9]. During the isolation process, a phase of characterizing phage properties should be included in the protocol, focused on determining the lytic capacity and spectrum of host bacteria. In some protocols, the use of chloroform is recommended for the extraction of phages. Currently, it is known that this inactivates enveloped phages, so it is not recommended; in the opposite case, the use of this organic compound would be helpful because the structure of enveloped phages has components of a lipid nature. With the isolation protocol, we consider obtaining the concentration that would allow obtaining a high titer of phage that has lytic capacity against a specific bacterial species [40].

However, there are exceptions, such as when phages are found in high titers in environmental samples; we can mix the sample with the host bacterial cell culture to achieve sufficient titers that generate inhibition halos during the characterization and phage lytic capacity. In this case, it is recommended to use a limited volume of environmental sample by plaque assay [46].

The phage concentration is generally low in most environmental samples, so adding an enrichment step to the isolation protocol is recommended. In the case of plants, the strategy for obtaining and isolating phages must be focused on taking samples from parts of the infected plant, such as leaves, roots, or stem or from irrigation water or soil [46]. 

### 5.1. Obtaining Bacteriophages from Environment Samples

The aim of the process is to find a phage with a high titer on media suitable for reproduction; for instance, tomato diseases have been treated with titers of 10^6^ to 10^8^ plaque-forming units (PFU)/mL [53]. The virus titer, including the phage titer, in seawater can be low, requiring preconcentration of the sample by filtration, precipitation, or both. Multiple studies recommend the use of mineral salts, such as zinc, calcium, or ferric chloride to concentrate phages from samples of seawater, wastewater, or any liquid in which the salt concentration can precipitate the phages, thus avoiding their being salvaged by water molecules, leading to phage precipitation [54,55,56].

On the other hand, bacteriophages in diluted samples can also be concentrated by flocculation, in the form of small insoluble aggregates (flocs) even at low phage concentrations. For wastewater samples, a low-speed centrifugation step is recommended to eliminate large insoluble parts and cells from the sample [57]. When using the filtration method, it is less common to concentrate phages from aqueous samples because the filters clog quickly. Nevertheless, it is an essential step in the phage collection protocol; most protocols end with a filtration step at 0.45 or 0.22 μm to eliminate bacteria, with the choice of filter pore size depending on the need to eliminate all bacteria. To retain very large bacteriophages, it is recommended to use a 0.45 μm membrane [58].

In summary, after sample collection, the phage titer can be evaluated directly, or concentrating it may be recommended, using a step of the protocol to achieve the necessary titer for the infection stage; this is achieved by directly applying the sample to the medium enriched with the host bacteria, considering that the sample must have a high enough titer to achieve successful infection. Another method that is commonly recommended is to add the phage enrichment step prior to infection by precipitation (ZnCl_2_ or CaCl_2_, or FeCl_3_) or floc and, in some cases, to use a filter to concentrate the phages and achieve a successful infection stage [9,40].

### 5.2. Experimental Detection of Bacteriophages

In this stage, it is recommended to use detection methods for new phage isolates, such as spot test, plaque test, or lysis in cultivation [59,60]. 

The spot test method consists of inoculating phages with the host bacteria, forming a lawn, and then placing droplets of phage on the plate. This incubation shows a lysis or halo effect related to phage activity. The advantage of this method is that it is simple and allows testing of multiple phages that are filtered on the same plate. A limitation of this method is that it requires growing the host in plate media. It is also prone to false positive results due to the lysis of bacteria by binding media components or by phages that do not lead to productive disease [43].

The plate test method consists of placing high phage dilutions obtained from filtrate together with the bacteria on the surface of the plate by extension or coating of soft agar. The plate is previously incubated with plaques to analyze afterwards. This shows evidence of phage growth, and the plaque indicates the lytic or lysogenic cycle. The size of the lytic plaque may indicate the size of the phage due to diffusion; a disadvantage is that the host must grow to converge on the plate. Most phages cannot be plated, even with highly productive hosts, due to limited agar diffusion [61]. In the culture lysis method, the phage filtrate is added to the bacterial culture broth and incubated for monitoring of cell lysis signals by the turbidity of the culture. Metabolic stains can also be used to measure the level of turbidity associated with metabolic activity. This method is used for bacteria that do not show confluence on the medium plate. Bacteria that grow in broth could be adapted for automation using spectrophotometry based on turbidity [59]. However, a limitation is the occurrence of false positives due to the absence of lysis. Cellular debris can inactivate phages by charge, affecting the infectivity. Hosts that evolve rapidly to become resistant to phages will cause false negatives [62].

In the routine dilution (RTD) method, phages are diluted to a titer that produces minor confluent cell lysis on a plate. This is used for phages that do not show morphological differences on plaques. Unfortunately, this method is inclined to produce false positives due to the fact that the media or components are not diluted [63]. The main challenges of phage detection are the inoculation quantity and specificity of the bacterial host and the viability and disposition of the phage for increasing the probability of successful infection that is possible to detect.

### 5.3. Detection of Bacteriophages from the Genome for Bacterial Biocontrol

Presently, there are repositories of sequenced genomes of organisms, and information on bacteriophages is available in NCBI’s PhagesDb and GenBank [60,64]. 

The genome is important in order to determine the constitution of a phage, and the host’s genotype and phenotype may be related. The phage genome is the genetic information that allows reproduction. Tailed dsDNA phages are the most studied, including tailed members of class *Caudoviricetes,* and families *Straboviridae*, and *Drexlerviridae* [31]. Other types of RNA and ssDNA phages make up a small group; there may be more that have not yet been discovered [10].

Another important characteristic is the size of the genetic material described, which can range from ~3300 nucleotides for ssRNA *E. coliphages* up to 500 kbp for *Bacillus megaterium* phage G. The size of dsDNA stem phage genomes ranges from ~11.5 kbp (Mycoplasma phage P1) to ~30 kbp (Pasteurella phage F108) for members of the families *Drexlerviridae*, *Autographiviridae*, and *Straboviridae* [31].

The phage genome encodes all the necessary components to generate new virions with structural proteins for the capsid and stem, and ligand proteins that cover the capsid, allowing it to interact with the host cell receptors and introduce genetic material. There may be other proteins that help in the development of infection, such as proteases, which help to evade the host’s immune response [10,42]. These genomes show the complexity of the microorganisms (bacteria and bacteriophages). In addition, we present an approach for comparative genome sequencing (Figure 2). Complete genomes of bacteriophages show high variability, which is a characteristic of bacteriophages. In these analyses, the sequence of phage Salvo was used as a template, and only phage Sano had a high conservation sequence >70% (yellow), because it belongs to its homologue; the other sequences added were phage phiXc10 (blue), *Ralstonia* phage RSM3 (pink), and *Agrobacterium* phage Atu_phe07 (green), which showed low conservation sequences between bacteriophage genomes (<30%) (Figure 2).

The sequences of genomes in the NCBI database showed that there are few sequenced genomes available. For the majority of bacteriophages, the genome is not available. Therefore, it is necessary to increase the number of genomes that can be obtained from other studies (assembly genomes), such as metagenomics studies, and to characterize bacteria and phage genome partners to understand host–phage relationships [65].

Currently, computational studies seek to complement the experimental laboratory studies carried out for the characterization of bacteriophages. Multiple bioinformatics applications have been developed for faster and better determination of candidates for phage therapy. Currently, the availability of viral genomic information has allowed the development of computational tools such as machine learning languages, for an approach called the phage classification tool set (PHACTS), which can identify the type of life cycle of a bacteriophage based on its protein sequences, because it is always required for the selection of lytic phages [66].

Another molecular implementation based on bacterial defense mechanisms is the use of mutations in receptors to prevent phages from interacting, making them unable to penetrate and inject their genetic material. Bacteria can detect regions in their genome that are susceptible to enzymatic cleavage of foreign DNA at specific sites. This is used as a molecular tool known as the Clustered Regularly Interspaced Short Palindromic Repeats/ Caspase 9 system (CRISPR/Cas system), which considers the immune system of bacteria that manage to confer resistance to phages [67].

Currently, genomic information is important for the development of analytical tools at the bioinformatics level. Leite et al. (2018) described an automated phage identification system from a genomic phage library that is capable of detecting effective phages from genomic information through a combination of machine learning and bioinformatics, in order to understand the phage–bacteria relationship by analyzing their genomes [67].

In this approach, a database is created in which the phage sequences are contained, and one for bacterial sequences is also created. Then, a model is created that is aimed at learning the specific characteristics of phages that interact with the bacteria, called positive interactions, and bacteria and phages are used for this probability model. Thus, there is non-experimental evidence that they have interacted, and negative interactions are obtained. Then, after selecting phage candidates based on their characteristics and positive interactions with the bacteria, they are analyzed from the deduced protein sequence at the genomic sequence level, with prediction of functional domains and protein interaction (phage–bacteria), where the focus is on membrane receptors, which is the information that feeds the machine learning model to select phage candidates from theoretical genomic and proteomic information [68].

Recently, Amgarten et al. (2018) developed a computational tool that predicts bacteriophage sequences from metagenomic data called Metagenomic Analysis and Retrieval of Viral Elements (MARVEL), which is based on machine learning. This program feeds on groups of 1247 phage sequences and 1029 bacterial genomes, determined from fragments of sequence counting that identifies sequences corresponding to semi-techniques of phages and bacteria with significant hits for viral proteins. Interestingly, in order to validate the operation and accuracy of MARVEL identification, the authors compared the results of the analysis of the same contigs with VIRsorter and VirFinder against MARVEL functional bases of software recommended to carry out the unknown virus identification. VIRsorter is based on alignments and searches for similarities in databases of known viruses, whereas VirFinder is based on a machine learning algorithm through K-mer frequency profiles that are obtained from the contigs and taken for training of the model [11,69,70]. The results of this study show that in the comparison of the three types of viral sequence analysis software, there were similarities except for two kbp fragments. It should be noted that MARVEL showed significant values (<0.001) for all cases analyzed with positive ranges. Therefore, this new computational tool will be able to support the design and study, as well as the identification, of phages from their genetic information [11].

A new approach, known as VIBRANT, focuses on the annotation of viruses. It is a new hybrid method that uses machine learning and protein similarity to determine genome quality and completeness, and characterizes viral communities from metagenomic assemblies. VIBRANT uses neural networks of protein signatures and newly developed v-score metrics to determine lytic viral genomes or prophages that are integrated. As a platform for evaluating viral community function, it was trained and validated with reference virus datasets, microbiomes, and virome data [71].

## 6. Bacterial Diseases Controlled by Bacteriophages 

Bacteria are prokaryotic microorganisms lacking a true nucleus. They have a cell wall composed of peptidoglycan, and their size is approximately 1 to 10 µm long and 0.5 to 2 µm wide [71]. Bacteria can be classified according to their phenotypic and genotypic characteristics. Phenotypic classification involves growth, cell morphology, texture, motility, pigmentation, and the formation of colonies [53]. Genotypic classification involves the DNA that constitutes the genome of the bacteria, which is plectonemic. This means that the DNA double helix undergoes coiling on the same molecule, giving rise to a higher-order helix [54]. Interestingly, the genome allows the classification to determine infectivity, pathogenicity, and, if present, prophage sequences, which can determine whether a group of bacteria will act as host for a specific phage [10].

Of particular interest are the gram-negative bacteria classified within the phylum Proteobacteria, which are phytopathogenic. Gram-negative bacteria have a complex outer membrane that contains a lipopolysaccharide, as well as structures known as porins that regulate the transport of molecules into and out of the cell. These bacteria contain a peptidoglycan layer approximately 7 to 8 nm thick located between the outer membrane and the cytoplasmic membrane. The lipopolysaccharide provides this type of bacteria with its extraordinary capacity to infect [65].

On the other hand, some genera can produce biofilm embedded in an exopolysaccharide matrix, making them resistant to antibiotics [72,73]. The size of the genome in gram-negative bacteria is estimated to be between 2.8 and 2.9 Kbp [74]. It is known that 40% of bacterial genomes are unique, but the function of bacterial genes is similar among species [74].

The infection mechanisms related to gram-negative bacteria are related to the content of glycans on the surface of cells, which participate in biosynthetic activities and cell wall regulation. These glycans include lipopolysaccharides, lipooligosaccharides, capsular polysaccharides, and N- and O-glycoproteins. They provide the ability to interact with host cells by host recognition of these macromolecules on the cell surface [75]. In plants, populations of 10^6^ colony-forming units are required to cause health problems; the damage is visible in the form of stains, mosaics, or rotting in leaves, fruits, and roots. Bacteria are disseminated through the soil, by the wind, or in wounds in the plant tissue or inoculated seeds. Humidity and temperature are essential factors in infection. *Pseudomonas syringae* pv. *phaseolicola* can cause disease below 22 °C (72 °F), and *Xanthomonas campestris* pv. *phaseoli* does so above 22 °C in beans (*Phaseolus vulgaris*) and can enter the plant through the stoma or via injection by phytophagous insects [40,50].

*Pseudomonas* bacteria cause damage to bean crops, manifested as round dark green spots on infected tissues, and their development requires high humidity conditions [76]. Tomato and other nightshade species are affected by *Ralstonia solanacearum*, which causes bacterial wilt; it also causes damage to crops, such as chili, potatoes, eggplants, tobacco, and 50 other plant families. It induces abnormal growth, wilting, and death, and its development is closely associated with relative humidity, along with high temperatures [77]. In the same way, the bacterium *Pectobacterium carotovorum* from the *Enterobacterales* order usually occurs in isolation or forms colonies, and favors temperatures between 4 and 38 °C. It is the cause of diseases related to rottenness in tomato, bean, pumpkin, peas, and chili. It spreads through wounds in the vegetable cuticle generated by external factors. The damage is reflected in the disintegration of pectins, and multiplies in the cells, generating soft areas causing the host plant’s death [72]. Strategies for the biocontrol of the different genera of proteobacteria with bacteriophages are described below.

### 6.1. Bacteriophages in the Biocontrol of Pseudomonas spp.

*Pseudomonas phaseolicola* is a pathogenic agent that has been studied for biocontrol using *Pseudomonas* phage phi*6*, which demonstrated possible bactericidal action against *P. phaseolicola* at 1 × 10^9^ PFU/mL. phi6 is classified as a cystovirus and has dsRNA. This bacteriophage infects *P. phaseolicola HB1OY* and can be obtained by enriched culture. Isolated *Pseudomonas* phage phi6 showed a constant adsorption rate of 3.3 × 10^10^ mL/min in a semi-synthetic medium and 3.8 × 10^10^ mL/min in a nutrient broth–yeast extract medium (Table 1) [78].

This biocontrol mechanism of bacteria was corroborated by characterization studies, with similar phages, *Pseudomonas* phages phi8, phi12, phi13, phi2954, phiNN, and phiYY, belonging to the same family. When used at 1 × 10^9^ PFU/mL, they showed effective control over *P. syringae*, which causes the initial foliar symptoms of bean brown spot. Brown spot refers to small water-soaked spots that develop into distinctive necrotic brown spots about 3–8 mm in diameter, often with narrow, diffuse yellow margins, and these lesions produce fall-out by damaging the leaves (Table 1) [23].

Similarly, studies on biocontrol of halo blight were carried out using lytic action phage F2 tested at 4 × 10^8^ on a bean crop. It showed efficiency up to 60%, suggesting its possible use in biological control techniques against *P. syringae* pv. *phaseolicola* (Table 1) [79].

In 2016, Addy and Wahyuni carried out characterization of phages *φSK2a*, *φSK2b*, φSK2c, and *φMGX1*, which showed the ability to generate lysis at 1 × 10^12^ PFU/mL in *P. syringae* pv. *glycinea*, which causes diseases in soybeans and other legumes (Table 1) [80].

**Table 1 ijms-24-00325-t001:** Application of bacteriophages as biocontrol agent against bacterial diseases in agricultural crops. Potential bacteriophages against infection by proteobacteria that cause diseases in bean, corn, sugar cane, green chili, tomato, tobacco, maize, potato, rice, pepper, geranium, sweet potato, onion, peach, cabbage, citrus, wheat, brassica, and passion fruit crops.

Bacterial Pathogen	Host Plant	Disease	Bacteriophages	Phage Therapy (PFU/mL)	Lysis (CFU/mL)	Genome (GenBank Number)	Reference
*Pseudomonas* *syringae* pv. *phaseolicola*	Bean (*Phaseolusvulgaris*)	Halo blight	Phage F2	4 × 10^8^	~10^8^	ND	[80]
			*Pseudomonas* phage phi8	1 × 10^10^	~2 × 10^8^	AF226851	[81]
			*Pseudomonas* phage phi6 *Pseudomonas* phage phi12 *Pseudomonas* phage phi13 *Pseudomonas* phage phi2954	10^7^ 1 × 10^9^	10^5^ 0.5 × 10^8^	M17461	[78,82,83]
						AF408636.1	
						AF261666.1	
						FJ608823.2	
			*Pseudomonas* phage phiNN	7.5 × 10^8^	~1.5 × 10^9^	KJ957164.1	[23]
			*Pseudomonas* phage phiYY	1 × 10^8^	1 × 10^9^	KX074201.1	[84,85]
*Pseudomonas* *syringae* pv. *syringae*	Wheat (*Triticum aestivum*)	Bacterial canker	*Pseudomonas* phage phi6	1 × 10^8^	3.9 × 10^8^	M17461	[86]
			Bacteriophage Phobos	2.5 × 10^6^	1 × 10^8^	MN478374.1	[24,87]
			φSK2a φSK2b φSK2c φMGX1	1 × 10^12^	1 × 10^8^	ND	[80]
			*Pseudomonas* phage MR1-MR8 MR12-M18	10^4^–10^7^	1.5– 5 × 10^8^	MT104465.1 MT104466.1 MT104467.1 MT104468.1 MT104469.1 MT104470.1 MT104471 MT104472 MT104473.1 MT104474 MT104475 MT104476 MT104477	[88]
*Xanthomonas* *axonopodis* pv. *phaseoli.*	Bean (*Phaseolus* *vulgaris*) Rice (*Oryza sativa*) Citrus (various species) Cassava root (*Manihot* *cassava*) Tomato (*Solanum lycopersicum*) Sugar cane (*Sacchrum offcicinarum*) Passion fruit (*Passiflora* spp.) Brassica (*Brassica* spp.)	Bacterial blight	CP2, ΦXac2005-1 ccΦ7 mccΦ13 ΦX.	5 × 10^9^	1 × 10^8^	ND	[89]
			ΦXaacA1	~10^6^	1 × 10^8^		
			Acm2004- ΦXacm2004-16 ΦX44	2.4 × 10^8^	1 × 10^8^		
			XacN1	1 × 10^10^	1 × 10^8^	ND	[90]
			Xcc9SH3	8 × 10^10^	1 × 10^8^	ND	[59]
			*Xanthomonas* phage Xaj2 *Xanthomonas* phage Xaj24	1 × 10^9^	1 × 10^8^	KU197014.1 KU197013.1	[91]
*Xanthomonas* *albileneans*	Sugar cane (*Sacchrum* *offcicinarum*)	Leaf scald	*Xanthomonas* phage phi Xc10	2.5 × 10^6^	1 × 10^8^	MF375456.1	[87,92]
			Phage Sano Phage Salvo *Xylella* phage Prado *Xylella* phage Paz	5–7 × 10^10^ 4 × 10^12^	1 × 10^8^	KF626665 KF626668.1 KF626667.1 KF626666.1	[93]
			Phage Cf2	2 × 10^9^	1 × 10^8^	ND	[94]
*Xanthomonas campestris* pv. *vesicatoria*	Green chili (Capsicum *annuum*)	Bacterial spot	Phage 1 Phage2 Phage3	4 × 10^8^ 3 × 10^8^ 7 × 10^8^	3 × 10^8^	ND	[25,95]
			ΦXaF18	6 × 10^10^	1 × 10^7^	ND	[96]
			*Xanthomonas* phage KΦ1 pXS	1 × 10^8^ 1 × 10^7^	1 × 10^8^ 1 × 10^8^	KY210139.1 ND	[52,97,98]
			AgriPhage	1 × 10^8^	1 × 10^8^	ND	[12,13,97]
			Phage 1 Phage 2 Phage 3	4 × 10^8^ 3 × 10^8^ 7 × 10^8^	3 × 10^8^	ND	[95]
*Xanthomonas campestri* pv. *campestri*	Cabbage (*Brassica* *oleracea*)	Cabbage rot	pXS	1 × 10^8^	1 × 10^8^	ND	[25,99]
			DB1	1 × 10^8^	1 × 10^8^	ND	[99]
*Xanthomonas* *campestri* pv. *pruni*	Peach (*Prunus persica*)	Bacterial spot	Xp3-A Xp3-I	4 × 10^9^	2 × 10^8^	ND	[100]
*Xanthomonas oryzae* pv. *oryzae*	Rice (*Oryza sativa*)	Leaf blight	φXOF4	1 × 10^8^	1 × 10^10^	ND	[101]
			*Xanthomonas* phage Xoo-sp2 Xoo-sp3 Xoo-sp4 Xoo-sp5 Xoo-sp6 Xoo-sp7 Xoo-sp8 Xoo-sp9	1 × 10^10^	~8 × 10^8^	KX241618.1 ND ND ND ND ND ND ND	[25,102]
*Xanthomonas* *axonopodis* pv. *allii*	Onion (*Allium cepa L.*)		Φ16 Φ17A Φ31	1 × 10^8^ 1 × 10^7^ 1 × 10^6^	1 × 10^8^	ND ND ND	[103]
*Pectobacterium* *carotovorum*	Corn (*Zea mays*)	Soft rot	ΦEcc2, ΦEcc3, ΦEcc9, ΦEcc14	1 × 10^7^	1 × 10^8^	ND ND ND ND	[104]
			*Pectobacterium* phage ZF40 ZF40-421	1 × 10^12^ 1 × 10^9^ 1 × 10^8^	1 × 10^8^	JQ177065.1 ND	[87,105]
			ZF40-RT80	1 ×10^8^	1 × 10^8^	ND	[106]
			POP72	5 × 10^6^	1 × 10^9^	ND	[107]
			ϕPD10.3			KM209229.1	[108]
			ϕPD23.1	1 × 10^5^	1 × 10^7^	KM209274.1	
			PP1	1 × 10^2^	1 × 10^4^	JQ837901.1	[107,109]
*Pectobacterium odoriferum*	Sweet potato (*Ipomoea batatas* *(L.) Lam*)	Bacterial root rot	Phi PccP-1	1 × 10^8^	1 × 10^8^	MW001769	[110]
*Ralstonia* *solanacearum*	Tomato (*Solanum* *lycopersicum*) Tobacco (*Nicotiana tabacum*) Geranium (genus) Potato (*Solanum* *tuberosum*) Banana (*Musa* *paradisiaca*)	Brown rot	ΦRSP	2.37 × 10^9^	2.1 × 10^8^	ND	[111]
			Qϕ-161	1 × 10^8^	1 × 10^8^	ND	[112]
			RsoP1EGY	1 × 10^8^	2 × 10^8^	NC_047946.1	[113]
			*Ralstonia* phage ϕRSL1	5.0	3 × 10^8^	NC_010811.2	[114]
			PE204	0.05	2 × 10^8^	ND	[115]
			*Ralstonia* phage ϕRSM3	6 × 10	1 × 10^8^	NC_011399.1	[26]
			ϕRSΒ1	6 × 10	1 × 10^8^	ND	[116]
			φSP1	1 × 10^6^	1 × 10^6^	ND	[117]
			*Ralstonia* phage RsoM1USA	1 × 10^8^	1 × 10^8^	M6752970	[118]
			*Ralstonia* phage RpY1	ND	ND	MN996301.1	[119]
*Burkholderia thailandensis* *B. caryophylli,* *B. gladioli* *B. glumae*	*Allium cepa* *Oryza sativa* *Nicotiana* *tabacum*	Onion skin Rotten rice grain blight Tobacco wilt	KS1, KS2, KS5, and KS6	1x10^7^	ND	ND	[120]
*B. plantarii*	*Oryza sativa*	Bacterial damping-off disease	FLC5	ND	ND	LC528882	[121]
*Dickeya* spp. *Dickeya solani*	*Solanum tuberosum*	Blackleg pathogen Soft rot	ϕD5	1 × 10^14^		NC019925	[108]
			Phage *myo* Phage *siph*	1 × 10^7^	3 × 10^9^	ND	[122]
			PP35	1 × 10^6^	1 × 10^6^	MG266157.1	[123]
			vB_DsoM_LIMEstone1 vB_DsoM_LIMEston 2	1 × 10^9^	ND	HE600015	[124]
*Clavibacter* *michiganensis*	Tomato (Solanum lycopersicum) Pepper (Capsicum annuum) Potato (Solanum tuberosum) Maize (*Zea mays*)	Bacterial canker Goss’s wilt	*Clavibacter* phage CMP1 *Clavibacter* phage CN77	4 × 10^7^	4 × 10^8^	GQ241246.1 GU097882.1	[125,126]
			CN8	5.3 × 10^6^	5.0 × 10^7^	ND	[127]
*Agrobacterium* *tumefaciens*	Tobacco (*Nicotiana tabacum*) Fruit tree (various species) Ornamental plants (various species) Forest trees (various species)	Crown gall disease	*Agrobacterium* phage Atu_ph07	1 × 10^5^	1 × 10^8^	MF403008.1	[48,128]
			*Agrobacterium* phage Atu_ph04 *Agrobacterium* phage Atu_ph08 *Agrobacterium* phage Atu_ph02 *Agrobacterium* phage Atu_ph03	1 × 10^8^	1 × 10^8^	MF403007.1 MF403009.1 NC_047845.1 NC_047846.1	[128]
			Milano	2.5 × 10^6^	1 × 10^8^	MK637516.1	[129]

For the control of bacterial brown spot, studies have been carried out using *Pseudomonas* phage phi6. The investigations showed an efficiency of 96.8 to 99% in reducing the disease, thus the phage was recommended for the control of *P. syringae* pv. *syringae* (Table 1) [86].

More recently, Amarillas et al. (2020) studied the Phobos phage (Table 1), a member of the *Siphoviridae* family with double-stranded DNA, and demonstrated its effectiveness when used at 2.5 × 10^6^ PFU/mL on 17 strains of *P. syringae.* It showed high lytic capacity of 64.7% against the disease caused by these bacteria [24]. Similarly, Rabiey et al. (2020) evaluated the effects of 13 phages as a means of controlling bacterial canker in cherry plantations in laboratory and greenhouse bioassays. The phages, which were used at 10^4^–10^7^ PFU/mL in the study, were *MR1, MR2, MR4, MR5, MR6, MR7, MR8, MR12, MR13, MR14, MR15, MR16*, and *MR18*, all *Caudovirales* belonging to the *Podoviridae*, *Myoviridae*, and *Siphoviridae*. The phages were applied to the leaves of infected plants, alone and in phage cocktails, and a general decrease was achieved in both experiments, from 15 to 40% for single phages to about 80% for the cocktail, which was reduced after five weeks. Thus, these phages could be recommended for use in the control of diseases caused by *P. syringae* pv. *morsprunorum*, and they may be equally effective in the control of *P. syringae* pv. *syringae*, which also causes diseases in various crops [88].

### 6.2. Bacteriophages in the Biocontrol of Xanthomonas spp.

Balogh et al. (2008) tested phage cocktails for treatment against *Xanthomonas axonopodis* pv. *phaseoli*, which produces cankers in oranges and grapefruits, including *X. axonopodis patovares citri* and *citrumelo*, in two experiments, one carried out in a greenhouse and the other in a commercial nursery. In the first experiment, phages *CP2*, *ΦXac2005-1, ccΦ7,* and *ccΦ13* were used at 5 × 10^9^ PFU/mL, and phage *ΦXaacA1* at 10^6^ PFU/mL. In the second study, a phage cocktail containing *ΦXacm2004-4, ΦXacm2004-16*, and *ΦX44* was tested at 2.4 × 10^8^ PFU/mL, alone, in combination with skim milk as an adjuvant, and in combination with mancozeb. The results indicated disease reduction from 48 to 59% when using phages alone, and no benefit was shown when phages were combined with skim milk or mancozeb (Table 1) [89].

Dömötör et al. (2016) carried out studies evaluating the control capacity of phages xaj2 and xaj24, on 35 strains of the bacterium *X. arborícola* pv. *juglandis*. These phages were effective at 1 × 10^9^ PFU/mL against at least 18 strains of the study bacteria with 88% effectiveness. The results indicated an additional control effect on pathovars of *X. campestris* and other species of bacteria, thereby confirming that they could be used as biological control methods against common bean rust (Table 1) [91].

The effects of bacteriophage *Xcc9SH3,* which has dsDNA and an icosahedral morphology and belongs to the *Siphoviridae* family, have been studied in vivo and in vitro; it has been shown to achieve high control over the *Xanthomonas* genus at a level of at least 8 × 10^10^ PFU/mL. Thus, it has been recommended as a control method to reduce the impact of bacteria that cause decay in various crops, such as cabbage and beans (Table 1) [58]. Phage *XacN1* has been studied as a method to control *Xanthomonas* spp. at 10^10^ PFU/mL, as well as *Xanthomonas citri*, which causes diseases in citrus (Table 1) [90]. Another study on the biocontrol of *Xanthomonas citri* with phage Cf2, a single-chain phage with lysogenic characteristics (at 2 × 10^9^ PFU/mL), reported a 43% decrease in the bacterial infection rate through a mechanism of coexistence with the host bacteria (Table 1) [94].

One study looked at the sequence genome of phage *Xanthomonas* phage Pagan, which was found to be able to infect Xanthomonas spp. in vitro. This phage was isolated from rice crop. The phage Pagan has a 62% GC and size of 44,448 bp genome [92].

Ahern et al. (2014) carried out studies with virulent phages Sano, Salvo, Prado, and Paz to prove their effectiveness in the control of *X. fastidiosa* and *Xanthomonas*. The phages have a 62.1% GC content, which is similar to the high values in the Xanthomonas spp., indicating that these bacteria could serve as hosts. All phages were tested at 4 × 10^12^ with *X. fastidiosa* strain Temecula at ~5–7 × 10^10^ PFU/mL for *Xantomonas* strain EC-12, and showed lytic effects on both bacteria, thus demonstrating that they could be used in the biocontrol of *X. fastidiosa* and *Xantomonas* spp. (Table 1) [93].

Other studies of pathovars were reported by Sadunishvili et al. (2015), who obtained positive results for the biocontrol of bacterial spot caused by *Xanthomonas vesicatoria* were tested three lines of phages (phages 1, 2, and 3) purified from six filtrates obtained from wastewater and tomato plants of regions in Georgia. The phages were tested on susceptible and resistant strains of *X. vesicatoria* on seedlings produced in the laboratory and in the greenhouse, and they achieved lytic effects in the range of 48 to 78%. In addition, the phages demonstrated polyvalent capacity and greater efficiency on resistant bacteria when used in a phage cocktail at 1 × 10^8^ PFU/mL or individual doses of 4 × 10^8^, 3 × 10^8^, and 7 × 10^8^ PFU/mL (Table 1). The phages showed stability at pH ranging from 6.8 to 7.2 and temperature between 55 and 82 °C, indicating that they could be used as a means of control for this disease [95].

Recently, Ríos-Sandoval et al. (2020) carried out genomic characterization studies of phage ΦXaF18, placing it within the *Myoviridae* family. Its lytic characteristics and control capacity against *Xanthomonas campestris* pv. *vesicatoria* were tested at 6 × 10^10^ on strains of bacteria that were inoculated with the bacteriophage and strains that were not. The results showed inhibited bacterial growth, thus suggesting that the phage could be used in phytosanity management programs (Table 1) [96]. In the same way, specific bacteriophages for bacterium *X. campestris* pv. have been used against *vesicatoria*, such as phages 1, 2, and 3 used at 4 × 10^8^, 3 × 10^8^, and 7 × 10^8^ PFU/mL, respectively, which were tested as a mixture and demonstrated great capacity for lytic infection, with values ranging from 86 to 91%. These phages have been located within the *Myoviridae* family; they show resistance to high temperatures, in the range of 50 to 70 °C, which may be ideal for field application (Table 1) [25].

Other studies have involved commercial phages for the treatment of *X. campestris* pv. *vesicatoria,* such as one by Obradovic (2005), who tested a mixture of specific bacteriophages in a product called AgriPhage (Table 1) [13]. The tests were carried out on tomato seedlings under greenhouse conditions, and a control effect on this bacterium was noted. When the commercial product Actigard was applied every 14 days on the seedlings, in combination with AgriPhage applied weekly, the lesions number per plant, which was 1.90 for untreated plants, was reduced. With Actigard + AgriPhage treatment, the number decreased to 0.27 lesions (Table 1) [13]. Actigard is a compound that induces systemic acquired resistance (SAR) in plants, initiates plant defense mechanisms against *X. campestris* pv. *vesicatoria* and prevents the occurrence of typical symptoms of tomato spot disease. AgriPhage is a mixture of six bacteriophage strains that are specific to *X campestris* pv. *vesicatoria* [13].

Recently, a list of bacteriophages used in the treatment of *Xanthomonas* ssp. was published by Stefani et al. (2021), which highlights the important aspects of biocontrol experiments performed with *Xanthomonas* spp. or panthovars. Experiments with phages KΦ1 (10^8^ PFU/mL), pXS (1 × 10^7^ PFU/mL), and AgriPhage (1 × 10^8^ PFU/mL) showed that they controlled *Xanthomonas euvesicatoria,* which causes bacterial spot in pepper. This was carried along with strains of arable microorganisms to generate biological control (*Bacillus subtilis*, *M. anisoplae*, *B. bassiana*) and a commercial fertilizer (Slavol), a copper-based agent combined with mancozeb and antibiotics (streptomycin sulfate and kasugamycin) [96]. These strategies show that a combination of established tools could increase the effectivity of bacteriophages as an alternative treatment for *Xanthomonas* diseases (Table 1) [96].

Finally, another review by Nakaryynga et al. (2021) noted that there are approaches for biocontrol of Xanthomonas spp. in greenhouse and field conditions [25]. In the early 19th century, Mallmann and Herstreet determined that a filtrate from rotting cabbage inhibited the growth of *X. campestri* pv. *campestri* in tissue. Since then, multiple phages have been investigated. Xp3-A and Xp3-I, used at 4 × 10^9^ PFU/mL, reduced infection by *X. pruni* in greenhouse conditions by 17–31% [101]. The cocktail of both phages applied immediately before applying the pathogen resulted in an inhibition of infection of 51–54% (Table 1) [100].

Other experiments have used a cocktail containing phages Φ16, Φ17A, and Φ31 at 10^8^, 10^7^, and 10^6^, respectively, which reduced *X. axonopodis* pv. *allii* infection in leaves by 43.3%, while monophage use of Φ31 reduced it by 26% compared with untreated controls (67.5%) at 9 days post-infection. This phage showed the broadest spectrum and lysed 12 strains of *X. axonopodis* pv. *allii* [103].

Phage φXOF4 inhibited the growth of *Xanthomonas oryzae* pv. *oryzae* at 1 × 10^8^ PFU/mL and no symptoms appeared, compared to symptoms in 73% of the untreated group [122,123]. Dong et al. demonstrated that applying premixed phage–pathogen suspension reduced infection development. That study used Xoo-sp2 and Xoo-sp9 at 10^10^ PFU/mL on 10 strains of *Xanthomonas oryzae* pv. *oryzae* (Table 1) [119].

Therefore, the use of phage cocktails is more effective against non-pathogenic Xanthomonas strains (npX, AXCB1201). It was also reported that phage (pXS, XcpSFC211) in broccoli plants reduced infection by *Xanthomonas campestris* pv. *campestris* by 18.9% compared to untreated plants, which developed infection at a rate of 93.7%, and pXS monotreatment at a rate of 86.2% (Table 1) [98]. Current studies are concerned with developing strategies (formulations) that can improve the effectivity of phages in the field and their UV protection to increase viability in infection zones and determine the lytic phages [25].

### 6.3. Bacteriophages in the Biocontrol of Pectobacterium spp. (Formerly Erwinia)

*Pectobacterium* spp. have been considered as pathogens that infect vegetables, causing soft rot. This genus presently is classified into *P. carotovorum*, *P. atrosepticum*, *P. aroidearym*, *P. aquaticum*, *P. betavasculorum, P. cacticidum*, *P. fontis*, *P. parmentieri*, *P. polonicum*, *P. polaris*, *P. peruviense*, *P. punijabense*, *P. wasabiae*, *P. zantedeschiae*, *P. versatile*, *P. odoriferum*, *P. brasiliense*, and *P. actibidiae*. Among them, *P. carotovorum* and *P. atrosepticum* are the most economically important bacterial plant pathogens [119].

The bacterium *Pectobacterium carotovorum* causes a plant pathology called bacterial soft rot in cane (Table 1) and is reported to be the agent responsible for rotting in calla lilies grown under greenhouse conditions. Ravensdale et al. (2007) reported that phages ΦEcc2, ΦEcc3, ΦEcc9, and ΦEcc14 at 1 × 10^7^ PFU/mL showed effective capacity in reducing the disease by 49 to 70% (Table 1) [104]. Other studies were carried out with phage *ZF40* in combination with its mutants, *ZF40-421* and *ZF40*, for the control of *P. carotovorum*; they were reported to present stability and had the ability to cause lytic effects on these bacteria [105]. Similarly, phage *ZF40-RT80* was reported to induce infection in *P. carotovorum* [106]. Phage *ZF40*, a member of the *Myoviridae* family, exhibits the ability to infect *Pectobacterium carotovorum* subsp. *carotovorum* at an average concentration of 1 × 10^12^ PFU/mL (Table 1) [105].

In addition, Voronina et al. (2019) reported on bacteriophage PP16 as a prospective agent for biocontrol of potato soft rot. This phage, which infected a broad range of *Pectobacterium carotovorum* strains, belongs the genus *Phimunavirus*, subfamily *Autographivirinae.* Bacteriophage PP16 inhibited the development of bacterial infection in in vitro experiments, and in plants the effect was determined with 10^6^ PFU/mL suspensions of PP16, resulting in a reliable four-fold decrease in affected areas compared to the positive control (Table 1) [130].

Recently, Shneider (2020) described the application of bacteriophage Arno160, which infects *Pectobacterium carotovorum*, as a potential lytic podovirus and a candidate for bacteriophage application for sort rot. This phage uses depolymerization of the O-polysaccharide lytic mechanism mediated by rhamnosidase [131]. Another bacteriophage that infects *Pectobacterium* strains is Jarilo, a novel genus of bacteriophages within the subfamily *Autographiviridae*, order *Caudovirales*. It was shown to infect 10 to 16 strains of *Pectobacterium atrosepticum* in tests reported by Perdersen et al. (2020) [132]. Jarilo presents a dsDNA genome of 40,557 bp with a GC content of 50.08% and 50 predicted open reading frames. This phage increased protection continuously for 1 h upon infection; however, there are no reports of bioassays in plants yet [131].

Recently, Lee et al. (2021) reported on the application of phage phiPccP-1 at 10^9^ to 10^10^ PFU/mL on infected *P. odoriferum* Pco14 using a spray on Kimbi cabbage leaves. The results showed a 95% reduction of damage in the leaves [110].

### 6.4. Bacteriophages in the Biocontrol of Ralstonia spp.

Studies have been carried out on the treatment and biocontrol of *Ralstonia solanacearum* with the use of the bacteriophage ΦRSP, a phage with a high capacity for infecting this bacterium. This phage, which presents lytic characteristics, is located within the *Myoviridae* family. It is a dsDNA bacteriophage that presents stability within a pH range of 4 to 11, can inhibit bacterial growth at 2.37 × 10^9^ PFU/mL, and can be applied in the field as a biological control method (Table 1) [111]. In rice cultivation research on control methods against *Xanthomonas oryzae* pv. *oryzae*, the causal agent of bacterial blight in this crop, the isolate phage *QΦ-16*, which belongs to the *Podoviridae* family, achieved control effects at 1 × 10^8^ PFU/mL. Its effectiveness was corroborated (Table 1) for *Xanthomonas campestris*, *Bacillus methylotropicus*, *Bacillus siamensis*, and *Ralstonia solanacearum*; the latter is a very representative pathogen of nightshades such as tomato (Table 1) [112].

Ahmad et al. (2018) carried out research in Egypt with a new double-stranded bacteriophage, RsoP1EGY phage, which is in the *Podoviridae* family. It presented control potential at 1 × 10^8^ PFU/mL on *Ralstonia solanacearum*. The studies were aimed at reducing the damage of this bacterium on potato crops, and the phage was tested on 20 strains of *R. solanacearum,* demonstrating specific control characteristics on at least 10 strains (Table 1) [113].

Similarly, studies were carried out to characterize the lytic *Ralstonia* phage phiRSL1, which has the ability to infect *R. solanacearum*, which causes diseases in tomato crops. It is classified as a member of the *Myoviridae* family, a type of phage called jumbo, with a genome of 240 kbp. It is capable of infecting bacterial strains of the *Ralstonia* genus at 5 PFU/mL under temperature conditions between 37 and 50 °C (Table 1) [114,133]. Phage PE204 has also been studied. This phage was isolated from pepper plants and has characteristics that make it a lytic phage. It belongs to the *Podoviridae* family and can survive at temperatures between 15 and 60 °C. Its effectiveness as pre-treatment at 0.05 PFU/mL as a means of controlling disease was tested. The results showed that it conferred healing capability on the diseased plants, which makes it a means of biological control for *Ralstonia solanacearum* (Table 1) [115].

Other studies carried out with the enveloped *Ralstonia* phage ϕRSM3, a member of the *Inoviridae* family, demonstrated effectiveness in preventing diseases caused by *R. solanacearum* when tomato plants were inoculated with cells containing the at 6 × 10 PFU/mL. During their development, treated plants did not show traits of the disease for a period of up to two months after treatment, in contrast to untreated seedlings, which died one week after the disease developed (Table 1) [26]. There are also data from studies carried out with phage φRSΒ1 at 6 × 10 PFU/mL. The results of the tests showed that it was capable of generating lytic infections on different strains of *R. solanacearum*, and characterization studies classified it as a dsDNA bacteriophage and a member of the *Podoviridae* family, which recommends it as a means of control over this type of bacteria (Table 1) [116].

Umrao et al. (2021) described phySp1, a bacteriophage that infected *R. solanacearum*, and characterization of the infection was tested in *Solanum lycopersium* (tomato) and *Solanum tuberosum* (potato). The bacteriophage required ~15 min for adsorption and the cycle life was 25–30 min by host cell lysis. Phage phiSp1 eliminated 94.73% of preformed *R. solacearum* biofilm at 10^6^ PFU/mL, and inhibited biofilm formation by 73.68% in vitro. Plant bioassays showed 81.39 and 87.75% reductions in potato and tomato disease, respectively (Table 1) [117]. In addition, *Ralstonia* jumbo phage RsoM2USA was isolated from the soil of infected tomatoes in the USA. This phage, a tailed member of the *Myoviridae* family, was able to infect 14 *Ralstonia* strains. Phage RsoM2USA reduced the infection of *Ralstonia solanacearum* RUN302 strain at 10^8^ PFU/mL and showed the ability to significantly reduce the virulence of *R. solanacearum* strain RUN302, *R. pseudosolanacearum*, and *R. syzyzii,* indicating that it could potentially be used as a biocontrol [134].

Recently, Lee et al. (2021) described the isolated *Ralstonia* phage RpY1 as a member of the *Podoviridae* family. It has a genome of 43,284 bp, a G+C content of 61.4%, and 53 open reading frames. This phage was capable of infecting three *Ralstonia* strains, SL341, LMG2300, and LMG2296, but there are no studies with evidence from biocontrol assays in plants [119].

### 6.5. Bacteriophages in the Biocontrol of Burkholderia spp.

*Burkholderia* spp. are associated with rhizosphere and soil and have a wide range of metabolic carbon sources. Many species are described as pathogenic and opportunistic in human and plant hosts. Principally the strains *B. caryophylli, B. gladioli*, *B. glumae*, and *B. plantarii* cause bacterial wilt in tobacco and iris flowers, and *B. gladioli* and *B. carnations*, or blight and rice rot. This kind of bacteria can be isolated from soil, water, and plants and rhizospheres include crops [135].

*Burkholderia cepacia* is a β-proteobacteria that is highly antibiotic resistant and have described nine strains that are genotypically different and phenotically similar. Seed et al. (2005) described the isolate of potential phages from sample soil, obtained ks1-ks11 bacteriophages that showed the effect of lysed on *B. gladioli* [135].

Jungkhun et al. (2018) reported that bacterial panicle blight (BPB) was caused by *B. glumae*, principally affecting rice. The authors described the isolates of phages NBP1-1, NBP4-7, and NBP4-8 lysing *B. glumae* BGLa14-8 strain. The foliar application NBPA-7 decreased the pathology by ≤ 62% [136].

Sasaki et al. described the isolate of a novel phytopathogenic *Burkholderia* phage obtained from leaf compost, named FLC5, which presented a genome of 32,090 bp of circular DNA. It was associated with *Podoviridae*, closer to *B. multivirans* phage KS14, and was able to infect *B. glumae* and *B. plantarii* (Table 1) [121]. Currently, more studies focused on *Burkholderia* spp. diseases in plants are needed.

### 6.6. Bacteriophages in the Biocontrol of Dickeya spp.

*Dickeya* spp. (formerly *Erwinia chrysanthemi* or *Pectobacterium chrysanthemum*) together with *Pectobacterium carotovorum* and *P. atrosepticum* (formerly *Erwinia carotovora* subsp. *carotovora* and *Erwinia carotovora* subsp. *atroseptica*) are bacteria responsible for economically relevant losses in potato and other crops worldwide [137].

Adriaenssens et al. (2011) described the characterization of vB_DsoM_LIMEstone1 and 2 bacteriophages, which have a T4 genome of 152,427 bp and a G+C content of 49.2%. The phages were shown to reduce rotting and blackleg in potato in laboratory experimental conditions, which means it could be used for the treatment of strains of *Dickeya solani* [124].

Czajkowski et al. described an isolate of ϕD5 bacteriophage belonging to the *Myoviridae* family, which presented a genome of 155,346 bp of dsDNA with GC content of 49.7%. This genomic information could be used to design strategies of biocontrol with this phage [108].

Recently, Kavanova et al. described a host-specific *Dickeya* bacteriophage, PP35, classified in the *Ackermannviridae* family, with a genome of 152,048 bp and GC content of 49.30%. It is able to infect and kill strains such as *D. solani*, and not *D. dianthicola*, *P. atrosepticum, P. parmentieri, P. carotovorum *subsp. *carotovorum*, and *P.c.* subsp. *Brasilense*. These qualities suggest that this phage may be a potential candidate for biocontrol [123].

### 6.7. Bacteriophages in the Biocontrol of Clavibacter michiganensis

Among the studies carried out on the biocontrol of diseases caused by *Clavibacter michiganensis*, Wittmann et al. (2010) and Wittmann et al. (2011) described phages CMP1 and CN77, which showed great specificity for the treatment of this bacteria at 4 × 10^7^ PFU/mL without causing secondary effects on beneficial soil bacteria [116,117]. Other studies on biocontrol of this pathogen were carried out with phage CN8 at 5.3 × 10^6^ PFU/mL, with the addition of polymers (polyvinyl alcohol), as a coating on corn seeds. This treatment was able to combat damage by *C. michiganensis* subsp. *nebraskensis* for up to four months (Table 1) [127].

### 6.8. Bacteriophages in the Biocontrol of Agrobacterium tumefaciens

Attai et al. (2018) investigated the effects of the application of *Agrobacterium* phage Atu_ph07, a member of the *Myoviridae* family and characterized as a jumbo phage, with lytic qualities and specific to *A. tumefaciens* at 1 × 10^5^ PFU/mL. It failed to infect other bacteria, thus recommending its use in crops affected by *A. tumefaciens* (Table 1) [48]. Similarly, Attai and Brown (2019) carried out studies to characterize new phages that were capable of exerting control over *A. tumefaciens*, and they found *Agrobacterium* phages Atu_ph04, Atu_ph08, Atu_ph02, and Atu_ph03, in the *Podoviridae* family. The results indicated that they are lytic phages at 1 × 10^8^ PFU/mL, with high specificity for lysis of *A. tumefaciens* without harming other soil microorganisms (Table 1) [128]. Finally, a recent study by Nittolo et al. (2019) described the phage Milano, from the *Myoviridae* family, with an 68,451 bp genome and a GC percentage of 52.5%, and with lytic and specific properties against *A. tumefaciens* C58, characterizing at as a phage with the ability to exert biocontrol over *A. tumefaciens* disease (Table 1) [129].

## 7. Conclusions and Perspectives

In the search for alternative treatments to improve the production of agricultural crops, here we show the current landscape regarding techniques to obtain phages for use in biocontrol. In summary, phage treatment has two stages: selection of the phage, followed by isolation, purification, and concentration, in vitro testing, and in vivo experiments using plants. Under normal conditions, it is possible to obtain a phage in approximately one to three months [60]. Therefore, we must consider the application of bioinformatics techniques as a complementary strategy to obtain phages from metagenomic samples, which could reduce the time needed for phage selection, thus reducing the number of phage candidates tested in the laboratory. The VIRsorter, VirFinder, MARVEL, VIBRANT, and RaFHA tools use genomic information to deduce the lytic capacity of phages and predict genes as lytic phage markers or host specificity, or predict viral proteins associated with important mechanisms of infection of the host by the phage [11,67,68,70,71,138].

These methods have been developed in recent years and have obtained significant result, in conjunction with databases giving annotations of the genomes, and together with experimental evidence. Taken together, all of this allows for improvements regarding algorithmic decisions, but a limitation is that for new phages and bacteria, it is more difficult to reach a conclusion using computational tools with the scant information available [10]. Computational tools are limited because the quantity of genome sequences for phages of agronomical interest deposited in databases need to be increased in order to improve the decisions and update the information on phage and host genomics used by bioinformatic tools.

Another aspect that should be studied is in-field application strategies for bacteriophages against plant diseases. This is a scenario where three organisms interact: bacteriophage, bacteria, and plant [53], and bacteriophages face an adverse microenvironment in which it initially depends on where the disease originates. This could be in the phyllosphere, or the aerial part of the leaves, where the bacteriophages will be in the presence of sunlight, and considering that bacteriophages are sensitive and generally inactive, it would be recommended to apply the phage dose during the evening [134].

In the case of the rhizosphere, from the bacteriophages’ perspective, it is an environment rich in bacteria and fungi, and the plant roots exchange materials such as phosphorus and nitrogen, which will enhance bacterial growth and require multiple phages. Among the factors that affect the viability of bacteriophages in this microenvironment are pH, type of soil, and different concentrations of mineral salts that affect phage–bacteria interactions and reduce the viability of phages, and thus their capacity to infect [53].

Multiple strategies for application have been tested to improve the effectiveness of bacteriophages and obtain optimal biocontrol of diseases in plants. These protocols widely recommend using a high phage titer between 10^4^ and 10^12^ PFU/mL, which can be applied to the soil or directly to foliage of infected plants in combination with adjuvants such as salt minerals or products such as AgriPhage [53,64,139].

Finally, plants have many problems due to bacterial infections, and the study and development of alternatives for the treatment of these diseases must be carried out by applying current bioinformatics technology and cutting-edge experimental methods that can increase our understanding of bacteriophage-bacteria-plant interactions to improve the development of phage therapy as a natural mechanism of bacterial biocontrol in agricultural crops.

## Figures and Tables

**Figure 1 ijms-24-00325-f001:**
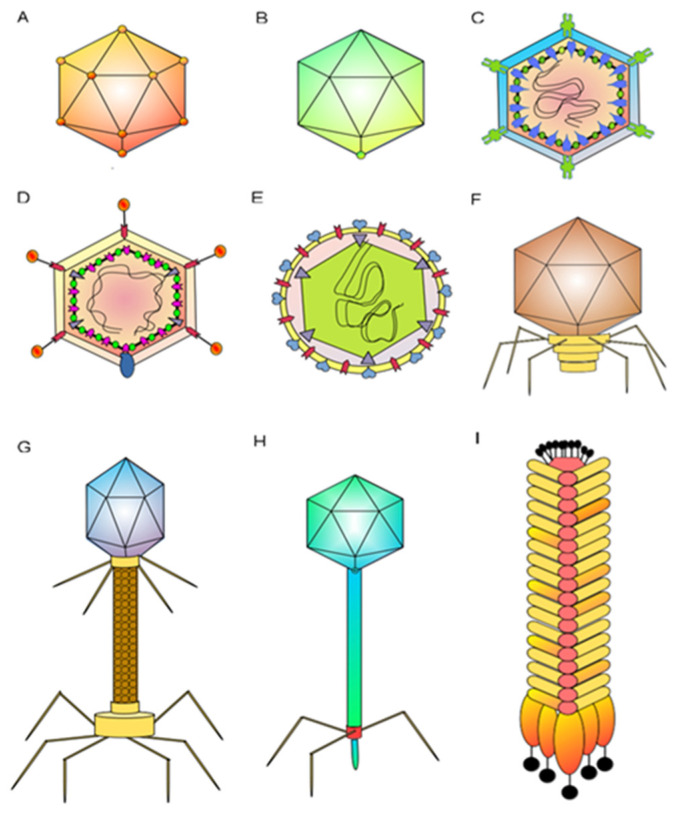
Schematic representation of the most commonly studied phages [16]: (**A**) levivirus MS2 has a capsid with icosahedral symmetry and a size of about 26 nm; (**B**) microvirus ϕX174 is a non-evolved icosahedral capsid about 30 nm in size; (**C**) podovirus T7 is a non-evolved icosahedral capsid about 60 nm in size; (**D**) tectivirus PRD1 is a non-enveloped icosahedral capsid with a size of about 66 nm; (**E**) cystovirus phi6 is an enveloped spherical virion 85 nm in diameter; outer and inner capsids have icosahedral symmetry; (**F**) corticovirusPM2 is an icosahedral capsid 56 nm in diameter; (**G**) myovirus T4 is non-enveloped, with a morphological head–tail structure about 110 nm in length; (**H**) siphovirus T5 is non-enveloped, with a head–tail structure; head is about 60 nm in diameter; and (**I**) inovirus M13 is non-enveloped, with rods of filaments 7 nm in diameter and 700 to 2000 nm in length.

**Figure 2 ijms-24-00325-f002:**
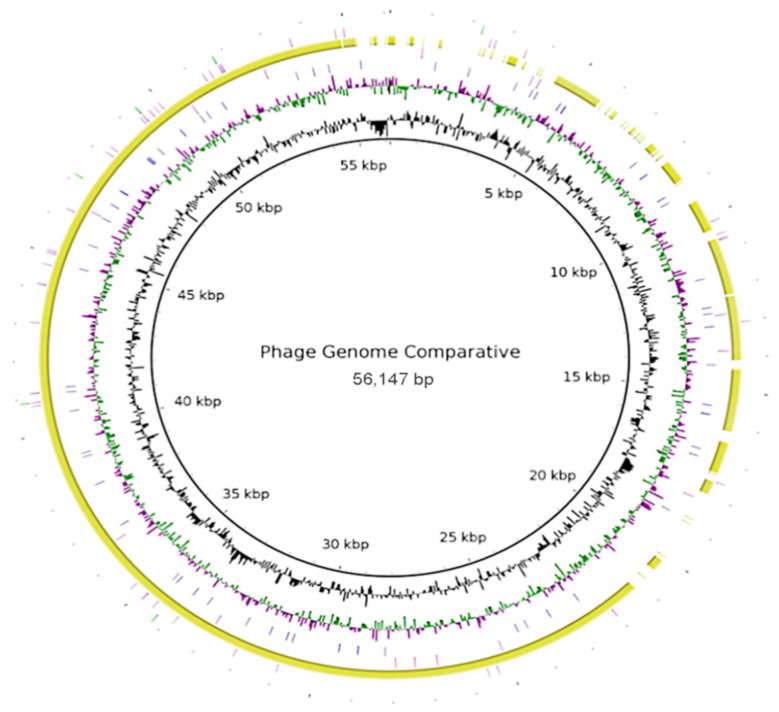
Comparison of bacteriophage genomes used for biocontrol of proteobacteria diseases present in agricultural crops. Complete genomes of bacteriophages were obtained from NCBI (Appendix A). Template sequence was phage Sano; % GC content (black) is shown. *Xanthomonas* phage phiXc10 (blue) and *Ralstonia* phage RSM3 (pink) show nucleotide sequence identity conservation <30%. Phage Salvo (yellow) shows identity about 70%, and *Agrobacterium* phage Atu_ph07 (green). Genomes were compared in Brig version 0.95 software, Brisbane, Australia (http://sourceforge.net/projects/brig/, accessed on 8 September 2022).

## Data Availability

The datasets generated during and/or analyzed during the current 331 study are available from the corresponding authors.

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
