# Peer review of "Phage Therapy for Crops: Concepts, Experimental and Bioinformatics Approaches to Direct Its Application"

_ijms, 2022, doi:10.3390/ijms24010325_

Round 1

Reviewer 1 Report

The authors have carried out a comprehensive review of the topic.

I apologise for the disheartening news, but the manuscript needs to be heavily and extensively edited by an individual that is familiar with English grammar, format, structure and style.

I would encourage the authors to spend the time in improving the manuscript.

Author Response

Thanks for the comments, and the manuscript was submitted to the English edition for improving it.

Reviewer 2 Report

The review of Matos-Pech et al., deals with an interesting subject that is gaining more spaces and applicability in agriculture. The way of which authors are describing the different approaches related to phage identification, isolation, characterization, and application in the field is rational, thus this illustrative information is worthy of being published. Before acceptance, authors should revise some English mistakes and report some minor corrections in the text, and hereafter are:

-Abstract: Replace with “Phage therapy consists of applying….

-Abstract: Replace with “alternative treatment against bacterial diseases.

-Abstract: Replace with “most relevant bacteriophages used for biocontrol of Pseudomonas……

-Introduction: reorder the chronological review of phages starting from the year of discovery (1917) and so on.

-Introduction: “responsible” not italicized.

- The title of paragraph 2 is very odd: please rephrase.

- The title of Figure1 is odd: eliminate Structure phages most studied. Please report the shape and approximative dimension of each phage\family in Figure 1, for a better understanding.

 -Figure 2: The first sentence is very odd, so please rephrase.

- The paragraph after Figure 2: should include a description on the % of nucleotides identity or variation present among phages of the same genera and\or families.

- Authors should include the demarcation criteria for the classification of phages (nothing is reported on that).

-Authors should add information on the life cycle (lytic, non-lytic) and mode of proliferation of phages.

- Paragraph 5: Change the title to read: Bacterial diseases controlled by bacteriophages.

- Figure 3: I didn’t catch what is the sense of this Figure…to say what? I would eliminate it.

Author Response

Dear Reviewer 2.

Thanks for the comments, and the following section shows the replies to each comment that was checked and performed each comment to improve the manuscript. Finally, the manuscript was submitted to the English edition.

Abstract: Replace with "Phage therapy consists of applying….

Reply line 16: the suggestion was added.

-Abstract: Replace with "alternative treatment against bacterial diseases.

Reply line 23: the suggestion was added.

-Abstract: Replace with "most relevant bacteriophages used for biocontrol of Pseudomonas……

Reply line 26: the suggestion was added.

-Introduction: reorder the chronological review of phages starting from the year of discovery (1917) and so on.

Reply lines 37-50: the suggestion was added.

-Introduction: "responsible" not italicized.

Reply line 40: the word was corrected.

- The title of paragraph 2 is very odd: please rephrase.

Reply line 81: the suggestion was added.

- The title of the Figure1 is odd: eliminate Structure phages most studied. Please report the shape and approximative dimension of each phage\family in Figure 1, for a better understanding.

Reply lines 97-106: the suggestion was added.

 -Figure 2: The first sentence is very odd, so please rephrase it.

Reply lines 386-387: the suggestion was added.

- The paragraph after Figure 2: should include a description of the % of nucleotide identity or variation present among phages of the same genera and\or families.

Reply lines 389-390: the suggestion was added: "show nucleotide sequence 389 identity conservation <30%. Phage Salvo (yellow) shows identity about 70%".

- Authors should include the demarcation criteria for the classification of phages (nothing is reported on that).

Reply lines 139-154: the suggestion was added, and we included a supplementary table about virus classification as a complement.

-Authors should add information on the life cycle (lytic, non-lytic) and mode of the proliferation of phages.

Reply lines 155-203: the suggestion was added, and we enrichment the text describing the mechanics of lytic and lysogenic cycles.

- Paragraph 5: Change the title to read: Bacterial diseases controlled by bacteriophages.

Reply lines 155-203: the suggestion was added.

- Figure 3: I didn't catch what is the sense of this Figure…to say what? I would eliminate it.

Reply: The figure was deleted.

Round 2

Reviewer 1 Report

This is a greatly improved version of the manuscript. I have enclosed my edits/suggestions in the PDF file.

I would encourage the authors to carefully check the entire manuscript, there are still many inconsistencies in scientific style.  Bacterial  genera are always in italics, and 'species" behind genus name is always spp. In addition - please note in bacterial nomenclature anything that follows spp. is never capitalised.

The bad news ....Since you have been working on this manuscript, phage taxonomy has been astronomically changed by ICTV. The Myoviridae, Podoviridae and Siphoviridae families have been eliminated and now we have phage genera solely based on genomics (you will have to consult the ICTV site or consult an phage taxonomist). Your sub-section on the phage taxonomy must be updated to reflect this new reality otherwise your manuscript is outdated before it is published. When describing morphology it is acceptable to use myovirus and podovirus (no capitals, no italics).

The phages in each of your subsections now belong to different taxa. I will leave it up to you as to how you want to deal with all the historically referenced work that mentions the old taxonomic designations. It would be a herrendous amount of work to check where some of these phage belong taxonomically.

Author Response

Dear reviewer 2.

Here will find your comments and replies highlighted in the color yellow.

For reviewer 2, the comments:

This is a greatly improved version of the manuscript. I have enclosed my edits/suggestions in the PDF file.

I would encourage the authors to carefully check the entire manuscript, there are still many inconsistencies in scientific style.  Bacterial  genera are always in italics, and 'species" behind genus name is always spp. In addition - please note in bacterial nomenclature anything that follows spp. is never capitalised.

The replies were found highlighted in yellow color on the manuscript.
Reply Thanks for your suggestion. The scientific style was revised in all manuscripts and homogenized and corrected based on your recommendations.

The bad news ....Since you have been working on this manuscript, phage taxonomy has been astronomically changed by ICTV. The Myoviridae, Podoviridae and Siphoviridae families have been eliminated and now we have phage genera solely based on genomics (you will have to consult the ICTV site or consult an phage taxonomist). Your sub-section on the phage taxonomy must be updated to reflect this new reality otherwise your manuscript is outdated before it is published. When describing morphology it is acceptable to use myovirus and podovirus (no capitals, no italics).

The phages in each of your subsections now belong to different taxa. I will leave it up to you as to how you want to deal with all the historically referenced work that mentions the old taxonomic designations. It would be a herrendous amount of work to check where some of these phage belong taxonomically.

Reply lines 142-172: Thanks newly for the suggestions. We revised the current literature about bacteriophages taxonomy and rewritten it. This section added the new references. In addition, indeed, the taxonomy based on genome viral is currently more accepted. However, a limit of this classification is that not all bacteriophages described are sequencing their genome, which only could be classified with the classic method. This review shows this case, where some bacteriophages are described and some, yet there is not genome sequencing. However, in this review, we do mention that currently performed computational analyses for reclassified the bacteriophages described in this review with the available genome.
Then, we justified the use of classic taxonomy because of the absence of genome sequence. In addition, the italics were revised and corrected in the manuscript and highlighted with yellow.

Thanks for your suggestions.

kind regards

Authors